# Artificial Sweeteners Disrupt Tight Junctions and Barrier Function in the Intestinal Epithelium through Activation of the Sweet Taste Receptor, T1R3

**DOI:** 10.3390/nu12061862

**Published:** 2020-06-22

**Authors:** Aparna Shil, Oluwatobi Olusanya, Zaynub Ghufoor, Benjamin Forson, Joanne Marks, Havovi Chichger

**Affiliations:** 1Biomedical Research Group, School of Life Sciences, Anglia Ruskin University, Cambridge CB1 1PT, UK; Aparna.Shil@pgr.anglia.ac.uk (A.S.); bf248@student.anglia.ac.uk (B.F.); 2London Epithelial Group, Department of Neuroscience, Physiology and Pharmacology, University College London, London NW3 2PF, UK; oluwatobi.olusanya.13@ucl.ac.uk (O.O.); Zay.Ghufoor@gmail.com (Z.G.); joanne.marks@ucl.ac.uk (J.M.)

**Keywords:** intestinal epithelium, permeability, sweeteners, claudin, Caco-2

## Abstract

The breakdown of the intestinal epithelial barrier and subsequent increase in intestinal permeability can lead to systemic inflammatory diseases and multiple-organ failure. Nutrition impacts the intestinal barrier, with dietary components such as gluten increasing permeability. Artificial sweeteners are increasingly consumed by the general public in a range of foods and drinks. The sweet taste receptor (T1R3) is activated by artificial sweeteners and has been identified in the intestine to play a role in incretin release and glucose transport; however, T1R3 has not been previously linked to intestinal permeability. Here, the intestinal epithelial cell line, Caco-2, was used to study the effect of commonly-consumed artificial sweeteners, sucralose, aspartame and saccharin, on permeability. At high concentrations, aspartame and saccharin were found to induce apoptosis and cell death in intestinal epithelial cells, while at low concentrations, sucralose and aspartame increased epithelial barrier permeability and down-regulated claudin 3 at the cell surface. T1R3 knockdown was found to attenuate these effects of artificial sweeteners. Aspartame induced reactive oxygen species (ROS) production to cause permeability and claudin 3 internalization, while sweetener-induced permeability and oxidative stress was rescued by the overexpression of claudin 3. Taken together, our findings demonstrate that the artificial sweeteners sucralose, aspartame, and saccharin exert a range of negative effects on the intestinal epithelium through the sweet taste receptor T1R3.

## 1. Introduction

Under normal conditions, the intestinal epithelial barrier maintains selective gut permeability to allow for nutrient absorption but provide a robust barrier and prevent the entry of pathogens and pathogenic molecules into circulation. The disruption of the intestinal epithelial barrier results in a ‘leaky gut,’ a key pathophysiological event seen in several chronic inflammatory disorders such as diabetes, pancreatitis, multiple organ failure, and autoimmune diseases [1,2,3]. The impairment of the intestinal barrier and the subsequent increase in leakage predominantly occurs in the distal small intestine and large intestine, and it can result in increased systemic inflammatory responses, tissue injury, and, in some severe cases, sepsis and increased mortality [4]. There are a variety of treatments for impaired gut permeability, including prebiotics and metformin [5,6]; however, there have been limited large-scale clinical trials to establish the efficacy of interventions in reducing permeability and its associated inflammation.

The intestinal barrier is maintained by two key mechanisms: epithelial cell homeostasis to regulate cell numbers forming the barrier and homotypic junctional complexes to regulate paracellular permeability across the barrier [7]. Intestinal epithelial homeostasis is established by equilibrium between cell proliferation and cell death, with dysregulated or excessive epithelial cell death associated with diseases of impaired barrier integrity and leakage across the paracellular space [4]. The paracellular space is largely modulated by tight junction (TJ) proteins that control the movement of water, nutrients, and electrolytes across the epithelium into the interstitial fluid [8]. TJ proteins are a family of around 50 proteins of which the main TJ proteins are occludins, claudins, and junctional-adhesion molecules (JAMs) [9]. Despite this, knockout studies have demonstrated a key role for claudins, rather than occludins and JAMs, in maintaining barrier integrity [10]. Some claudins, e.g., claudins 1, 3, 4, 5, 8, 11, and 14, can form homotypic or heterotypic interactions at the epithelial cell–cell junction to form a seal and reduce leakage, and some claudins, e.g., claudins 2, 7, 10, and 15, form a pore to increase leak [9,11,12]. Expression of claudins at the TJ can be regulated via protein kinase (PKC)-mediated phosphorylation and downstream intracellular trafficking pathways, such as reduced lysosomal degradation or increased trafficking to the TJ complex at the epithelial cell surface, to influence barrier integrity [13,14,15,16]. As such, diseases associated with a leaky gut, such as Crohn’s disease and ulcerative colitis, have been demonstrated to impact the expression of claudins in the small and large intestine; however, the mechanism for this is not clear [16,17]. Studies are therefore required to understand the mechanisms that regulate the expression and function of claudins in the intestinal epithelium and how these proteins may be involved in controlling leakage across the gut.

There is increasing evidence that the consumption of a high-fat diet and excessive alcohol intake leads to intestinal permeability and metabolic endotoxemia [18,19,20]. These changes in permeability are associated with increased oxidative stress and the reorganization of claudin expression at the tight junction [21,22]. While a link between the Western diet and intestinal permeability has been established, the effect of food additives found in the diet is not well-understood. A range of artificial sweeteners are increasingly being utilized as non-caloric sugar substitutes, of which aspartame, sucralose, and saccharin are most commonly consumed in both food and beverages [23,24]. There is controversy regarding the metabolic effects of these acutely-sweet molecules, with studies demonstrating both a positive and negative role for artificial sweeteners in the diet [25,26,27,28]. Studies in humans and mice have demonstrated that the consumption of artificial sweeteners in the diet is linked with the dysbiosis of gut microbiota and an associated increase in levels of endotoxins secreted from these bacteria, such as lipopolysaccharide (LPS) [27,28]. LPS released from the gut microbiota is linked to an increase in intestinal permeability [29]; however, while changes in the gut microbiota are a potential regulator of this permeability, the direct effect of artificial sweeteners on intestinal permeability is not well-understood. 

In the study presented here, we used the well-established intestinal epithelial cell line, Caco-2, to study the effect of the commonly-consumed artificial sweeteners aspartame, sucralose, and saccharin on intestinal epithelial cell viability, monolayer permeability, claudin expression, and oxidative stress response. This study utilized a variety of concentrations that could be achieved in the diet to address the impact of these sweeteners on the intestinal epithelium. As artificial sweeteners stimulate the sweet taste receptors T1R2 and T1R3 to elicit a taste response pathway [30], we also sought to demonstrate the role of these G protein-coupled receptors in regulating the effect of aspartame, sucralose, and saccharin on the intestinal epithelium. We anticipate that findings from this study could expand our understanding of the physiological effect of artificial sweeteners and indicate the effect of dietary components on gut health.

## 2. Materials and Methods

### 2.1. Cell Lines and Reagents

Human colon carcinoma cells (Caco-2) were purchased from Sigma-Aldrich (Dorset, UK), cultured in Eagle’s Minimum Essential Media containing 10% fetal bovine serum and 1% penicillin/streptomycin, and used between passages 35 and 50. Silencing RNA (siRNA) and a DharmaFECT^™^ reagent were obtained from Dharmacon (Cambridge, UK). Claudin 3 and control vector cDNA were purchased from GenScript (Piscataway, USA). A Lipofectamine 3000^™^ reagent was purchased from Thermo Fisher Scientific (Paisley, UK). DCFDA (2′,7′-dichlorofluorescin diacetate) and antibodies directed against claudins 2, 3, 4, and 7 were purchased from Abcam (Cambridge, UK), while a claudin 15 antibody was obtained from Novus Biologicals (Abingdon, UK) and T1R3 and actin antibodies were obtained from Santa Cruz Biotechnology (Santa Cruz, CA). An annexin V kit was purchased from BD Pharmingen (Wokingham, UK). All other reagents, including the artificial sweeteners saccharin, sucralose, and aspartame, were purchased from Sigma Aldrich (Dorset, UK).

### 2.2. Animals and Ethics

The study used six male C57BL/6 mice bred at the Comparative Biology Unit at UCL’s Royal Free Campus. Animals were allowed ad libitum access to water and a standard rat chow (Diet RM1, SDS Ltd., Witham, Essex, UK) until the time of experimentation. Animals were housed in groups of 3–4 and were maintained on 12-h light–dark cycling (7a.m.–7p.m.) at a temperature of 15–22 °C. At 8–10 weeks, mice were anaesthetized with an intraperitoneal injection of 60 mg.kg^−1^ of pentobarbitone sodium (Pentoject, Animalcare Ltd., York, UK), and the monitoring of the pedal and corneal reflex was undertaken to ensure that deep anesthesia was achieved before the small and large intestine were removed. Euthanasia was performed by cervical dislocation, with death confirmed by the cessation of the heartbeat. The gut was separated into the following sections: duodenum (beginning at the stomach and ending at the ligament of Treitz), jejunum (from the ligament of Treitz until halfway along the small intestine), ileum (remaining half of the small intestine until the caecum), proximal (1st half of the colon), and distal colon (2nd half of the colon). Each segment was flushed with ice cold 0.9% saline to remove any contents and transferred to a cold glass surface at 4 ºC and cut open. The mucosa was scraped off using a glass slide and placed in 1 ml of RNAlater, snap frozen, and stored at −80 °C until use. All procedures were carried out in accordance with the UK Animals (Scientific Procedures) Act, 1986, Amendment Regulations 2012. The protocols were approved by the University College London (Royal Free Campus) Comparative Biology Unit Animal Welfare and Ethical Review Body (AWERB) committee. 

### 2.3. RT-PCR 

Total RNA was extracted from mucosal scrapes of the distinct regions of the mouse small and large intestine, or from Caco 2 cells, using a Trizol reagent (Life Technologies, Paisley, UK) according to the manufacturer’s instructions. RNA was DNase treated and reverse transcribed using a Transcriptor First Strand cDNA Synthesis Kit (Roche Diagnostics, Mannheim, Germany). A negative control was included for each sample. Claudin, taste receptor. and β-actin transcripts were analyzed by real-time PCR using a Fast Start Essential Green Master kit (Roche Diagnostics, Mannheim, Germany) on a Roche LightCycler 96 (Roche Diagnostics, East Sussex, UK) using the specific primers shown in Table 1 and Table 2. For all primers, the cycling conditions were as follows: 95 °C for 10 min followed by 40 cycles of 95 °C for 10 sec, 60 °C for 10 sec, and 72 °C for 10 sec. Relative gene expression compared to β-actin was established using the LightCycler 96 software. 

### 2.4. Annexin V Assay

Caco-2 cells were grown to 60% confluence in T-25 flasks prior to exposure with artificial sweeteners (range of concentrations) for 24 h. Artificial sweeteners were dissolved in the vehicle control (H_2_O) and sterile-filtered to prepare a working stock solution. Adhered and floating cells were collected and incubated with a binding buffer, annexin V, and propidium iodide for 15 min in the dark. Cells were then analyzed with an Accuri C6 Flow cytometer (BD Biosciences), and the percentage of positive cells for annexin V and propidium iodide was calculated with FlowJo (V10.2, Oregon, USA). 

### 2.5. siRNA and cDNA Transfections

Caco-2 cells were transiently transfected with siRNA specific to T1R3 or non-specific control siRNA using a DharmaFect^™^ 2 reagent, as per manufacturer’s guidelines. Alternatively, Caco-2 cells were transiently transfected with wild-type human claudin 3 cDNA (clone ID: OHu26411, vector: pcDNA3.1^+^/C-(K)DYK, or DYK control vector cDNA) using Lipofectamine 3000^™^, as per manufacturer’s guidelines. Cells were transfected at a seeding density of 0.5 × 10^4^ cells per well of a 96-well plate, 2.5 × 10^4^ cells per well of a Transwell insert, or 1.5 × 10^5^ cells per well of a 6-well plate. Transfected cells were plated onto Transwell inserts or 96-well plates for an analysis of permeability and whole-cell ELISA, respectively. At 24 h post-transfection, cells were exposed to artificial sweetener (100 µM) or vehicle control (H_2_O) in the presence and absence of the positive control LPS (1 µg/mL) for a further 24 h. Experiments were then performed as outlined in ‘permeability’ and ‘whole-cell ELISA.’ To confirm knockdown of T1R3 or the overexpression of claudin 3, at 48 h post-transfection, cells were lysed with a radioimmunoprecipitation assay buffer, resuspended in a Laemmli buffer, and subjected to immunoblot analysis. Immunoblot analyses were performed on 10% SDS-PAGE using a primary antibody specific to T1R3, the DYK tag, and β-actin at a dilution of 1:1000 and secondary antibody dilutions of 1:5000. 

### 2.6. ROS Assay

Caco-2 cells (1 × 10^4^ cells per well) were plated on black-walled 96-well plate for 24 h, followed by exposure to the cell permeant, fluorogenic dye DCFDA (10 µM), or dimethyl sulfoxide (DMSO) control for 30 min at 37 °C in the dark. DCFDA was then removed and replaced with artificial sweeteners (range of concentrations), the vehicle control (H_2_O), or the positive control LPS (1 µg/mL) for 1.5 h. DCFDA fluorescence was measured at 488 nm using a fluorescent plate reader (Victor, Perkin Elmer), and measurements were compared to cells cultured in the absence of DCFDA. 

### 2.7. Whole Cell ELISA

Caco-2 cells (1 × 10^4^ cells per well) were plated on black-walled 96-well plate for 24 h, followed by exposure to artificial sweeteners (range of concentrations), the vehicle control (H_2_O), or the positive control LPS (1 µg/mL) for a further 24 h. Where stated, cells were first transfected with siRNA or also exposed to N-acetyl cysteine (NAC) (1 mM) or the vehicle for NAC (H_2_O). Cells were then rinsed once with Dulbecco’s phosphate-buffered saline (DPBS) and fixed using 1% paraformaldehyde at room temperature for 10 min. Whole cell ELISA was then performed as previously described [34] in non-permeabilized Caco-2 cells using antibodies specific to claudins 2 (ab76032), 3 (ab214487), 4 (ab210796), 7 (ab207300), and 15 (NBP1-59267). Antibodies were specific to the αα 150 region of claudins [35]. Fluorescent-conjugated secondary antibodies were measured at a 1 sec exposure time using a florescent plate reader (Victor, Perkin Elmer), and measurements from blank wells (no primary antibody) were subtracted to provide the presented data.

### 2.8. Epithelial Monolayer Permeability

Epithelial monolayer permeability was assessed using the fluorescein isothiocyanate (FITC)-dextran permeability assay and validated with transepithelial resistance (TER) (EVOM2; World Precision Instruments, Herts, UK). For the analysis of monolayer permeability, Caco-2 cells were plated onto Transwell filters for 24 h, followed by exposure to artificial sweeteners (range of concentrations), the vehicle control (H_2_O), or the positive control LPS (1 µg/mL) for a further 24 h. Where stated, cells were first transfected with siRNA or exposed to NAC (1 mM) or the vehicle for NAC (H_2_O). Permeability was measured by adding FITC-conjugated to 20 kDa dextran (FD20) to media in the upper chamber of the Transwell filter to a concentration of 5 µg/µl. FD20 was allowed to equilibrate for 180 sec at 37 ºC, and a sample (100 µl) of media from the lower chamber was collected and analyzed at 488 nm using a fluorescent plate reader (Victor, Perkin Elmer). Permeability (%) was calculated by fluorescence accumulated in the lower chamber divided by fluorescence in the upper chamber, which was then multiplied by 100. 

### 2.9. Cell Viability and Morphology Studies

Caco-2 cell viability was assessed using the Cell Counting Kit-8 (CCK-8). Cells were exposed to artificial sweeteners or the vehicle control (H_2_O) for 24 h, followed by incubation with the CCK-8 reagent for 2 h at 37 °C. Absorbance was then assessed at 450 nm using a microplate reader (Tecan Sunrise), and viability was calculated as % normalized to vehicle. 

### 2.10. Statistical Analysis

The experimental number is presented in the legend for each experiment. In vitro experiments with Caco-2 cells were performed in duplicates. Data were analyzed using GraphPad Prism 7.0. For two groups, the variance in data sets was analyzed using the Mann–Whitney test followed by the T-test. For three or more groups, variance was assessed by using Bartlett’s test with data sets not reaching significance studied by Kruskal–Wallis test followed by Dunn’s test. For all other data sets, differences among the means were tested for significance in all experiments by ANOVA with Tukey’s range significance difference test. Significance was reached when *p* < 0.05. Values are presented as mean ± standard error mean (S.E.M.).

## 3. Results

### 3.1. High Physiological Concentrations of Artificial Sweeteners Decrease Viability and Increase Apoptosis of Caco2 Cells through the Sweet Taste Receptor (T1R3)

Artificial sweeteners stimulate the sweet-taste receptors T1R2 and T1R3, which are G protein-coupled receptors [30]. We and others have demonstrated the expression of T1R2 and T1R3 protein and mRNA in the intestinal epithelium where they have been identified to act as sensors to stimulate glucose absorption and modulate incretin release [36,37,38]. Given the wide range of concentrations of different artificial sweeteners consumed in the diet [23], we sought to understand the dose-dependent effect of the commonly consumed artificial sweeteners sucralose, aspartame, and saccharin on Caco-2 cell viability, apoptosis, and cell death. Caco-2 cell viability was significantly decreased by aspartame and saccharin at concentrations of ≥1000 µM (Figure 1a). Interestingly, there was a significant increase in cell viability following exposure to 1000 µM sucralose; however, at higher concentrations (10,000 µM), the sweeteners decreased cell viability (Figure 1a). Whilst there have been limited studies to indicate the concentration of sweeteners found in the intestine, following the consumption of artificial sweeteners, this range of concentrations (1–10 mM) is potentially achievable in the intestine and thus physiologically-relevant for members of the general population who regularly consume significant amounts of artificially sweetened foods. For example, a single chewing gum contains 0.01 mM, one can of soft drink contains up to 2 mM of artificial sweetener, and, more generally, the main additives in a range of products including diet drinks, sports drinks, snacks, and confectionary are artificial sweeteners [23]. Given that the acceptable daily intake for these sweeteners is high (between 14 and 40 mg/kg of body weight), it is likely that the public can consume high quantities of sweetener in the diet to achieve up to 10 mM exposure to sweeteners [23]. Artificial sweeteners have been established to bind to the sweet taste receptors T1R2 and T1R3; therefore, we next sought to establish the mRNA expression and cell surface protein levels of T1R2 and T1R3 in Caco-2 cells. Both the mRNA (ratio: 1.48 × 10^−6^ ± 1.11 × 10^−7^) and protein (83.84 ± 2.13 r.f.u.) expressions of T1R3 were identified in untreated Caco-2 cells; however, T1R2 mRNA was not detected in the cells (undetected), and only a low abundance of the protein was detected at the cell surface (13.69 ± 0.33 r.f.u.). We therefore next studied whether artificial sweeteners affected cell viability through T1R3 using the siRNA knockdown of the sweet taste receptor (63.5 ± 2.7% decrease, *p* < 0.05, *n* = 6); see Figure 1b. The significant decrease in cell viability following exposure to sucralose, saccharin, and aspartame at 10,000 µM was abolished by T1R3 knockdown (Figure 1c). These findings were supported by experiments with propidium iodide and annexin V staining in Caco-2 cells exposed to 0–1000 µM artificial sweeteners to measure cell death and apoptosis, respectively. A significant increase in cell death was observed at 1000 µM saccharin and aspartame (Figure 1d), matched by an increase in apoptosis at 10 and 100 µM (Figure 1e). As for cell viability studies, sucralose had no impact on Caco-2 cell death or apoptosis at concentrations of ≤1000 µM (Figure 1d,e). These findings demonstrated that saccharin and aspartame induce apoptosis at lower concentrations (up to 100 µM) and cell death at higher concentrations (≤1000 µM). Taken together, these findings indicated that, at high but physiologically-relevant concentrations in the small intestine, artificial sweeteners sucralose, aspartame, and saccharin decrease cell viability by binding to the sweet taste receptor T1R3. The findings also demonstrated a differential effect of aspartame and saccharin versus sucralose on Caco-2 cell apoptosis and death.

### 3.2. Low Physiological Concentrations of Artificial Sweeteners Sucralose and Aspartame Disrupt the Intestinal Epithelial Barrier through the Sweet Taste Receptor

Given the detrimental effect of high concentrations (≥1000 µM) of artificial sweeteners on Caco-2 cell viability, we next sought to establish the impact of a lower concentration (100 µM) on intestinal barrier function. The bacterial endotoxin LPS has been demonstrated to increase permeability of the intestinal epithelium [39]. Indeed, we demonstrated an increased epithelial monolayer permeability when using both 1 and 10 µg/mL of LPS using an FITC-dextran permeability assay (% permeability—1 µg/mL: 185 ± 6.9%, *p* < 0.05 versus vehicle; 10 µg/mL: 213.7 ± 10.7%, *p* < 0.05 versus vehicle, *n* = 6) and TER measurements (1 µg/mL: −263 ± 9.8 ohms, *p* < 0.05 versus vehicle; 10 µg/mL: −304 ± 15.2 ohms, *p* < 0.05 versus vehicle, *n* = 6). Caco-2 cell viability, however, was only decreased at 10 µg/mL of LPS exposure (% viability—10 µg/mL: 70 ± 1.5%, *p* < 0.05 versus vehicle, *n* = 6)). Therefore, 1 µg/mL of LPS was used as a positive control for subsequent experiments. The exposure of Caco-2 cells to the artificial sweeteners sucralose and aspartame significantly increased the permeability of the epithelial barrier to a similar level as seen for LPS (Figure 2a). Conversely, saccharin had no effect on epithelial barrier integrity (Figure 2a). Interestingly, the siRNA knockdown of the sweet taste receptor T1R3 attenuated sucralose- and aspartame-induced permeability but had no impact on LPS-induced leakage across the epithelial barrier (Figure 2b). These findings demonstrated the effect of the artificial sweeteners, sucralose, and aspartame on intestinal epithelial barrier function.

### 3.3. Sucralose and Aspartame Modulate Claudin 3 and 15 Expression in Intestinal Epithelial Cells through T1R3

Given the effect of sucralose and aspartame on epithelial barrier function, we next sought to establish the mechanisms regulating this process. Claudins are a key component of the tight junction complex and regulate epithelial barrier integrity. Though 26 claudins have been identified [40], in keeping with other studies [32,41], we demonstrated the expression of claudins 2, 3, 4, 7, 8, 15, and 23 using the RT-PCR analysis of the small and large intestine of mice (Figure 3a). From this data, we chose the most abundant claudins and determined their expression levels in our Caco-2 cell model. Interestingly, claudin 4 was most abundantly expressed in Caco-2 cells, while claudin 7 was almost undetectable in the cells in comparison to the high levels present in the murine intestine (Figure 3b). 

Claudins regulate permeability when expressed at the tight junction of the epithelial cell surface, with claudins 2 and 15 associated with pore formation and leakage and claudins 3, 4, and 7 linked to tight junction sealing and reduced leakage [40]. Therefore, we next studied the effect of sucralose and aspartame on the cell surface protein expression of claudins, using LPS as a control to mimic the breakdown of the epithelial barrier. Similar to mRNA expression, claudin 2 protein levels were found at low levels at the cell surface, with no significant effect of LPS on the protein (Figure 4a). In contrast, LPS exposure resulted in a significant decrease in claudin 3, 4, and 7 and a significant increase in claudin 15 protein expression at the cell surface (Figure 4b–e). Interestingly, both sucralose and aspartame significantly decreased claudin 3 surface expression and increased claudin 15 surface expression, similar to LPS exposure (Figure 4b,e), while the sweeteners had no effect on claudin 4 and 7 expression at the epithelial cell surface (Figure 4c,d). To assess the role of the sweet taste receptor T1R3 in regulating the effect of sucralose and aspartame on claudin 3 and 15 expression, these experiments were repeated in Caco-2 cells with T1R3 siRNA. The knockdown of T1R3 levels significantly abrogated the sweetener-induced decrease in claudin 3 surface expression but had no impact on the increase in claudin 15 levels (Figure 4f,g). Interestingly, the LPS-mediated decrease in claudin 3 was unaffected by T1R3 knockdown (Figure 4f). 

Taken together, these data highlight the differential expression of claudins in the murine intestine and cultured intestinal epithelial cells, and they demonstrate a key role for sucralose and aspartame in regulating the expression of claudin 3 and 15 at the epithelial cell surface. The data further indicated that sweeteners modulate claudin 3 and 15 expression at the tight junction through T1R3-dependent and T1R3-independent signaling pathways, respectively. 

### 3.4. Overexpression of Claudin 3 Rescues Sweetener-Induced Barrier Leakage Across the Intestinal Epithelium

To confirm that sucralose and aspartame regulate barrier leakage across the intestinal epithelium through claudin 3, our next experiments were performed in Caco-2 cells overexpressing wild-type CLDN3-DYK cDNA, or the vector control (DYK). Western blotting and whole-cell ELISA confirmed the overexpression of the construct at protein and cell surface levels, respectively (Figure 5a,b). The overexpression of claudin 3 had no impact on Caco-2 cell viability (Figure 5c), indicating no negative side effects of the transfection on the cells. Interestingly, leakage across the intestinal epithelial cell monolayer, induced by aspartame and sucralose, was abrogated by claudin 3 overexpression (Figure 5d). These data demonstrated a key role for claudin 3 in regulating the sweetener-induced permeability of the intestinal epithelium. 

### 3.5. Aspartame, But Not Sucralose, Increases Oxidative Stress in Intestinal Epithelial Cells Linked to Barrier Leakage

Finally, we sought to establish the mechanism through which the sweet taste receptor and the artificial sweeteners sucralose and aspartame regulate claudin 3 expression at the cell surface. Oxidative stress is an important regulator of claudin 3 localization in the intestinal epithelium and is linked to LPS-induced permeability [42,43], so we studied the effect of the artificial sweeteners sucralose and aspartame on the production of reactive oxygen species (ROS) in intestinal epithelial cells, using LPS as a positive control. The exposure of Caco-2 cells to LPS and aspartame, but not sucralose, significantly increased ROS production (Figure 6a). Interestingly, aspartame-induced ROS production was attenuated by exposure to the antioxidant NAC (Figure 6b) and the knockdown of T1R3 (Figure 6c). We next studied the role that oxidative stress plays on the sweetener-induced permeability of the Caco-2 cell monolayer and claudin 3 surface expression. Whilst NAC significantly attenuated aspartame-induced monolayer permeability, the antioxidant had no effect on sucralose-mediated leakage (Figure 6d). Similarly, the reduction in claudin 3 expression at the cell surface, induced by aspartame, was abrogated by NAC (Figure 6e), while sucralose-induced claudin 3 downregulation was unaffected by NAC (Figure 6e). Interestingly, aspartame-induced ROS production was significantly attenuated by the overexpression of wild-type claudin 3 (Figure 6f), thus indicating a reciprocal relationship between oxidative stress and claudin 3. 

Taken together, these data demonstrate that aspartame, but not sucralose, mediates claudin 3 expression at the tight junction and increases permeability of the epithelium through the production of ROS. The data further demonstrate a role for T1R3 in regulating aspartame-induced ROS accumulation in the intestinal epithelial cell. 

## 4. Discussion

At present, a large proportion of the population consumes artificial sweeteners, primarily aspartame, sucralose, and saccharin [23]; however, there is significant controversy regarding the impact that artificial sweeteners in the diet exert on health. In particular, the effect of sweeteners on both the diversity and function of the gut microbiota, a key factor that regulates intestinal permeability, has been previously established with associated metabolic disruption linked to this dysbiosis [27,28]. However, whether there is a direct effect of these sweet-taste molecules on intestinal permeability is not well-understood. In the present study, we demonstrated the effect of the artificial sweeteners, saccharin, sucralose, and aspartame on intestinal epithelial cell claudin expression, barrier integrity and ROS production. Our experiments showed the detrimental and differential effects of these non-nutritive sweeteners at concentrations that would be typically found in the diet. Findings from this study also further our understanding of the mechanisms that regulate the permeability of the intestinal epithelium and contribute to the controversy regarding the use of artificial sweeteners in the diet.

The intestinal epithelial barrier is vital in maintaining selective permeability between the small and large intestine, as well as circulation. The integrity of this barrier is maintained, in part, through cell survival, with an increase in epithelial cell apoptosis resulting in permeability, both in vitro and in vivo [44]. Tight junctions comprise another mechanism that regulates epithelial permeability, in particular the localization of claudins in the tight junction complex. In this study, we identified the expression of claudins 3, 4, 7, and 15 in the murine small and large intestine and in Caco-2 cells; however, only claudin 3 and 15 were downregulated and upregulated, respectively, in response to sucralose and aspartame treatment. Previous studies have demonstrated that dietary components, such as gluten, alter claudin 3 and 15 expression [45,46]; however, this is the first study to indicate that artificial sweeteners regulate these tight junction proteins. We further demonstrated the importance of claudin 3, rather than claudin 15, in regulating sweetener-induced permeability through T1R3. This may not be surprising given that claudin 3 is a barrier-sealing tight junction protein that is down-regulated in settings of intestinal permeability, while the pore-forming claudin 15 is associated with mucosal differentiation in the small intestine [47,48]. Interestingly, our experiments also demonstrated a role for claudin 3 in regulating aspartame-induced oxidative stress in the intestinal epithelial cell. Previous in vivo studies have demonstrated a role for oxidative stress in the dysregulation of claudin 1, 2, and 4 expression at the tight junction due to reduced levels of the antioxidant and superoxide dismutase or increased levels of hypoxia-inducible factor-1 [49,50]. In addition, in gastric epithelial cells, claudin 3 was identified to be sensitive to oxidative stress, with the siRNA knockdown of the tight junction protein exacerbating the permeability of the monolayer [51]. Our experiments indicated T1R3 as a key regulator of claudin 3-associated oxidative stress and the monolayer permeability of the intestinal epithelial barrier. 

Claudin expression is maintained through coordinated cell signaling processes in the intestinal epithelial cell. The shuttling of claudin proteins to the epithelial cell surface, to form the TJ complex, is dynamically regulated through intracellular trafficking processes [52]. Our experiments demonstrated that the artificial sweeteners aspartame and sucralose bind T1R3 to cause the reduced cell surface expression of claudin 3. The internalization of claudin 3, associated with the disruption of the TJ, has been observed to be caveolin- and flotillin-dependent [53,54]. It is therefore possible that downstream T1R3 signaling promotes these trafficking molecules to increase claudin 3 internalization; however, further studies are needed to establish this mechanism. Furthermore, the phosphorylation of TJ proteins, including claudin 3, by PKCζ has been demonstrated to play a key role in maintaining the TJ complex and therefore barrier function in the intestine [14,55]. Whilst the link between PKCζ and sweet taste sensing is not yet known, T1R3 stimulation by aspartame and sucralose may inhibit this PKC isoform and therefore promote the disruption of the TJ. Finally, β-catenin, Forkhead box O4 (FOXO4), and hepatocyte nuclear factor alpha bind to claudin promoters to regulate the expression of the TJ proteins; therefore, sucralose and aspartame may affect claudin 3 levels by blocking these transcription factors to reduce expression in the intestinal epithelium [56,57,58]. Further studies are needed to understand the molecular mechanisms through which sweeteners reduce claudin 3 expression at the TJ, as well as the resulting downstream effects on barrier permeability and ROS production. 

The expression of the sweet taste receptors T1R2 and T1R3 has been established in the intestinal epithelium [26,36,38]. Similar to a study by O’Brien and Corpe [59], data from the present study demonstrate the expression of T1R3, but not T1R2, in Caco-2 cells. Artificial sweeteners have been demonstrated to bind to the sweet taste receptor in extra-oral locations to regulate a range of processes including glucose transport and insulin secretion [38,60]. In the present study, we demonstrated that the artificial sweeteners saccharin and aspartame exert a toxic effect on intestinal epithelial cells at high concentrations, sucralose and aspartame increase epithelial permeability, and only aspartame causes oxidative stress. These findings are in contrast to previous findings where saccharin, but not aspartame or sucralose, was demonstrated to disrupt epithelial barrier integrity [61]. This difference in findings may have been due to the short time point of 3.5 h studied by Santos et al. as opposed to the 24 h time point assessed in the current experiment. This time-dependent difference in tight junction formation and permeability has been observed in other epithelial tissues. In the choroidal plexus epithelium, a comparison of acute (three hours) versus chronic (20 h) exposure to phorbol myristate acetate demonstrates alternate effects on paracellular permeability [62]. This form of ‘claudin-switching’ has also been observed in the intestinal epithelium of mice in settings of obesity [21]. Therefore, it is possible that the impact of aspartame and sucralose on the intestinal epithelium, in the short term, is minimal due to tight junction compensation, while a delayed, more comprehensive rearrangement of claudin proteins causes the detrimental response which we note at 24 h. This highlights the complex and dynamic nature of the claudin organization and barrier function of the epithelium, and it indicates that artificial sweeteners may have different effects on the intestinal epithelium in short and long term studies. 

In vivo studies have demonstrated that a saccharin-enriched diet causes the accumulation of water in the stool of rats which may be indicative of leakage across the intestinal epithelium [63]. Therefore, while further in vivo studies are needed to demonstrate the exact cellular effects of sucralose, saccharin, and aspartame on leakage across the intestinal epithelium, the in vivo experiments performed by Anderson and Kirkland over a 10-day period matched our in vitro findings at 24 h with regards to saccharin. However, it is worth noting that these experiments utilized exceedingly high concentrations of saccharin (>250 mM). Furthermore, whilst sucralose and saccharin are resistant to hydrolysis in the small intestine, aspartame is hydrolyzed into aspartic acid, methanol, and phenylalanine [64,65,66]. Though peak levels of hydrolysis products are observed in the plasma of humans at one-to-two hours post-consumption, it takes up to 24 h for levels to become undetectable [67,68,69]. Aminopeptidase A, the enzyme that is key for aspartame hydrolysis [64], is predominantly expressed and active in the mid and distal regions of the small intestine [70]. Proximal sections of the small intestine, the duodenum, and early jejunum may therefore be exposed to unmetabolized aspartame, which is able to bind to the sweet taste receptor, T1R3, expressed in these regions [71]. In contrast, the late jejunum and ileum are more likely to be exposed to the hydrolysis products of aspartame that do not bind to T1R3. It is worth noting that findings from the present study used an in vitro model of the intestinal epithelium, so further assessments with vivo experiments is necessary to confirm the physiological relevance of these findings. 

Our experiments did, however, show that T1R3 is key to the observed cellular effects, with the knockdown of the receptor attenuating the permeability, decreased viability, and ROS production induced by sweeteners. While these findings need to be confirmed using an in vivo permeability model to establish the physiological relevance of consuming sweeteners at these concentrations, previous studies have demonstrated that results from Caco-2 cell culture closely correlate with in vivo measurements of permeability [44]. The differential effects may have been a result of altered intracellular signaling downstream of T1R3 for each artificial sweetener. These differences may have been due, in part, to the structure of the artificial sweeteners, with the region of ligand binding in T1R3 mediating the signaling response. However, further studies are needed to identify the specific characteristics of each sweetener, their binding affinity to the sweet taste receptor T1R3, and the resulting effect on intracellular signaling. 

Artificial sweeteners are consumed by the general public at a range of concentrations, depending on the perceived sweet taste of the molecule and dietary choices, with one can of soft drink containing between 0.5 and 2 mM of sweetener [23]. A study of new food products launched in the USA between 1999 and 2004 showed that sucralose and aspartame are two of the most commonly-used artificial sweeteners, while saccharin is typically found in food and drinks blended with aspartame [23,72]. Our experiments established a detrimental effect of sucralose, saccharin, and aspartame on cell viability at a concentration of 10 mM, which, although high, is physiologically-achievable and within the acceptable daily intake given the increasing consumption of these sweeteners in both food and drinks [23,73]. Indeed, many studies in the field have utilized concentrations of up to 10 mM of artificial sweeteners [38,61]. Interestingly, at the significantly lower concentration of sucralose and aspartame (0.1 mM), we observed leakage across the intestinal epithelial barrier. These in vitro findings indicate that both low and high amounts of these two sweeteners disrupt intestinal epithelial cells. 

## 5. Conclusions

In the intestinal epithelial cell line, Caco-2, the commonly-consumed artificial sweeteners, sucralose, aspartame and saccharin, all have an impact on permeability. At high concentrations, aspartame and saccharin induce apoptosis and cell death, while at low concentrations, sucralose and aspartame increased epithelial barrier permeability and down-regulated claudin 3 at the cell surface. Whilst findings demonstrate that the artificial sweeteners sucralose, aspartame, and saccharin exert a range of negative effects on the intestinal epithelium through the sweet taste receptor T1R3, further study is needed to demonstrate the in vivo effect of these sweeteners on the intestinal epithelium.

## Figures and Tables

**Figure 1 nutrients-12-01862-f001:**
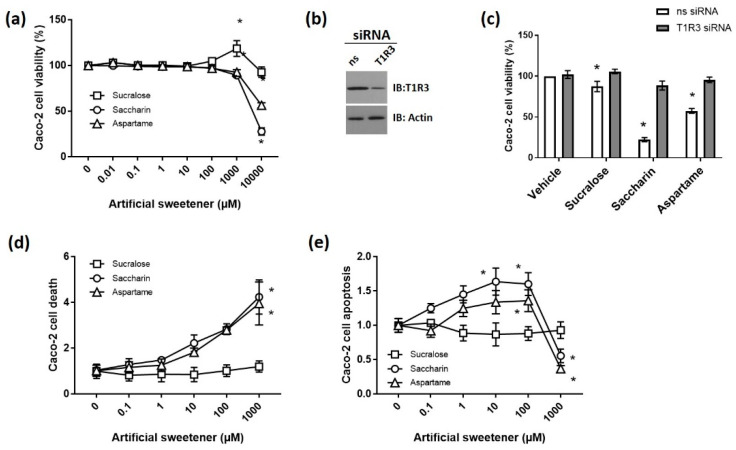
High physiological concentrations of artificial sweeteners decreased the viability and increase apoptosis of Caco2 cells through the sweet taste receptor (T1R3). (**a**) Caco-2 cell viability was measured using the Cell Counting Kit-8 (CCK8) assay following 24 h exposure to the artificial sweeteners sucralose, saccharin, and aspartame at concentrations ranging from 0.01 to 10,000 µM. Absorbance was normalized to a 0 µM control and expressed as percentage viability. *n* = 8. (**b**,**c**) Caco-2 cell viability was measured as for (**a**) following the siRNA knockdown of T1R3 for 24 h and exposure to sucralose, saccharin, and aspartame (10 mM) for a further 24 h. A representative blot of T1R3 and load control actin were shown to confirm siRNA knockdown using 50 µg of protein (**b**). Caco-2 cells were collected following exposure to sucralose, saccharin, and aspartame at concentrations ranging from 0.1 to 1000 µM for 24 h and analyzed by flow cytometry. Cell death (**d**) and apoptosis (**e**) were measured as propidium iodide and annexin V-positive and annexin V-positive cells, respectively. *n* = 5–6. Data are expressed as mean ± S.E.M. * *p* < 0.05 versus vehicle (0 µM).

**Figure 2 nutrients-12-01862-f002:**
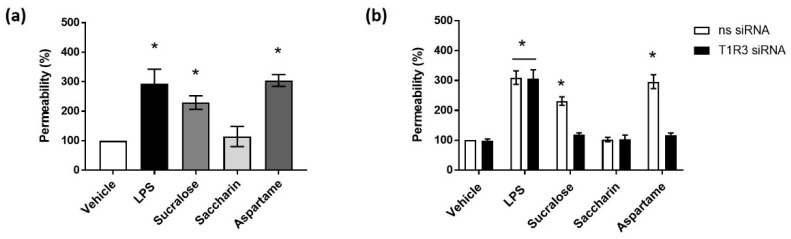
Low physiological concentrations of artificial sweeteners sucralose and aspartame disrupt the intestinal epithelial barrier through the sweet taste receptor. (**a**) The permeability of the epithelial monolayer was measured with an FITC-dextran assay, following exposure to sucralose, saccharin, and aspartame (0.1 mM) for 24 h using lipopolysaccharide (LPS) (1 µg/mL) as a positive control. (**b**) The permeability of the Caco-2 cell monolayer was measured with an FITC-dextran assay following the siRNA knockdown of T1R3 for 24 h and exposure to sucralose, saccharin, and aspartame (0.1 mM) for a further 24 h. % permeability was calculated normalized to vehicle treatment. *n* = 6. Data are expressed as mean ± S.E.M. * *p* < 0.05 versus vehicle (0 µM).

**Figure 3 nutrients-12-01862-f003:**
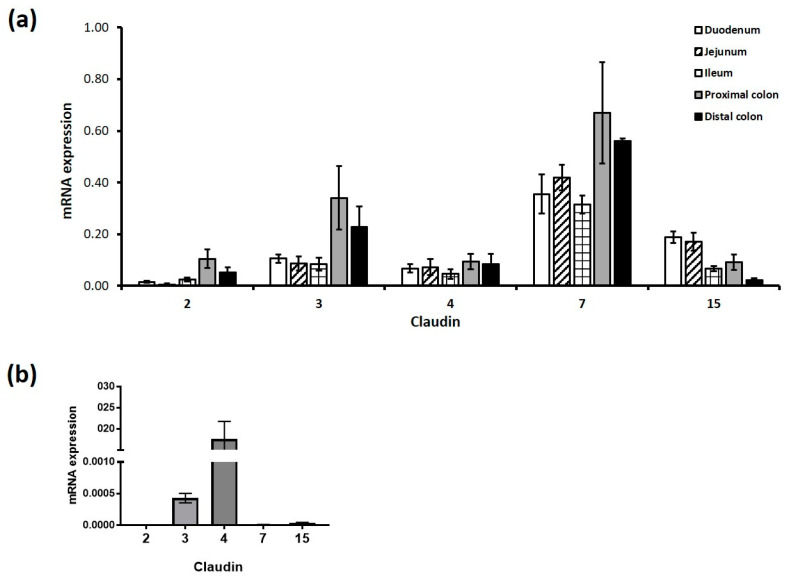
Claudin mRNA expression profile in the intestinal epithelium in vitro and in vivo. Claudin mRNA transcripts were profiled in the small and large intestine and in cultured Caco-2 cells. (**a**) Three segments of the murine small intestine, the duodenum (stomach to ligament of Treitz), the jejunum (ligament of Treitz until mid-small intestine), the ileum (remaining half of small intestine until the caecum), and two segments of the large intestine (proximal (first half of the colon) and distal colon (second half of the colon)) were collected for RT-PCR analysis. (**b**) Untreated Caco-2 cells were collected for RT-PCR analysis. The relative ratio was calculated as claudin compared to actin mRNA expression levels. Data are expressed as the mean ± S.E.M of PCR reactions. *n* = 6.

**Figure 4 nutrients-12-01862-f004:**
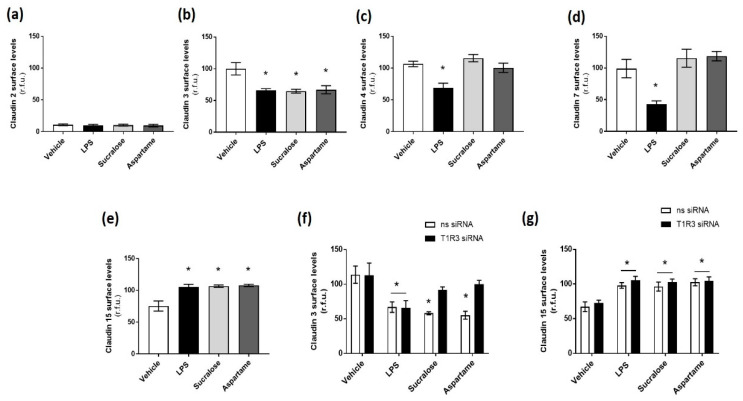
Sucralose and aspartame modulate claudin 3 and 15 expression in intestinal epithelial cells through T1R3. (**a**–**e**) The cell surface expression of claudins 2, 3, 4, 7, and 15, respectively, was determined, with whole-cell indirect ELISA using chemiluminescence, in Caco-2 cells exposed to sucralose or aspartame (0.1 mM) or LPS (1 µg/mL) for 24 h. (**f**,**g**) The cell surface expression of claudins 3 and 15, respectively, was determined in Caco-2 cells following siRNA knockdown of T1R3 for 24 h and exposure to sucralose or aspartame (0.1 mM) for a further 24 h. *n* = 6. Data are expressed as mean ± S.E.M. * *p* < 0.05 versus vehicle (0 µM).

**Figure 5 nutrients-12-01862-f005:**
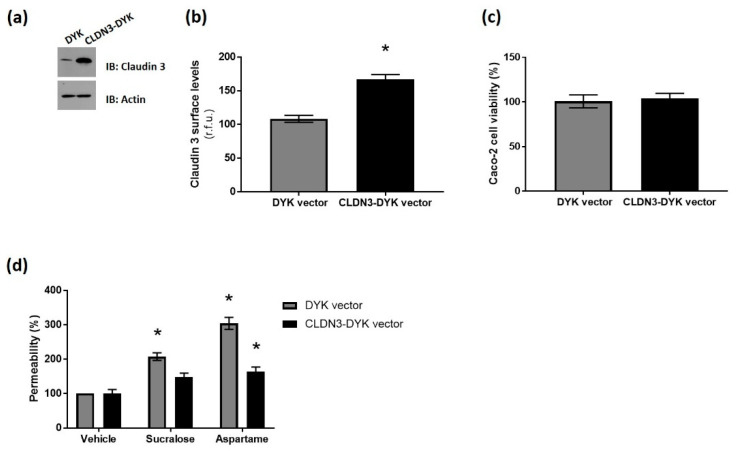
Overexpression of claudin 3 rescues sweetener-induced barrier leakage across the intestinal epithelium. Caco-2 cells were transiently transfected with cDNA encoding wild-type claudin 3 (CLDN3-DYK) or the vector control (DYK), or untransfected (UT) for 48 h. (**a**) The total protein levels of claudin 3 in cells were measured by the Western blot analysis of cell lysates using 50 µg protein. (**b**) The cell surface expression of claudin 3 was determined by whole-cell indirect ELISA using chemiluminescence in transfected Caco-2 cells. (**c**) The viability of transfected Caco-2 cells was assessed by a CCK8 assay. Absorbance values were normalized to untransfected control and expressed as percentage viability. (**d**) The permeability of the transfected Caco-2 cell monolayer was measured by an FITC-dextran assay following exposure to sucralose and aspartame (0.1 mM) for 24 h. % permeability was calculated normalized to vehicle treatment. *n* = 6. Data are expressed as mean ± S.E.M. * *p* < 0.05 versus untransfected cells or vehicle (0 µM).

**Figure 6 nutrients-12-01862-f006:**
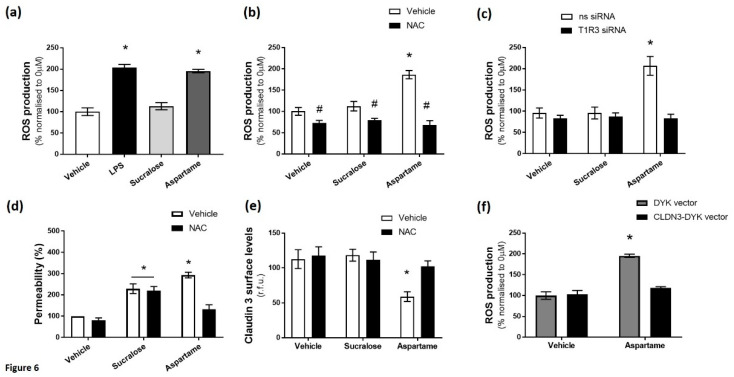
Aspartame, but not sucralose, increases oxidative stress in intestinal epithelial cells linked to barrier leakage. Reactive oxygen species (ROS) production in Caco-2 cells was measured by fluorescence of DCFDA following exposure to sucralose or aspartame (0.1 mM) or LPS (1 µg/mL) as a positive control (**a**) in the presence and absence of the anti-oxidant N-acetyl cysteine (NAC; 1 mM) (**b**) or following siRNA knockdown of T1R3 (**c**). (**d**) The permeability of the Caco-2 cell monolayer was measured by an FITC-dextran assay, following exposure to sucralose and aspartame (0.1 mM) for 24 h in the presence and absence of NAC. % permeability was calculated normalized to vehicle treatment. (**e**) The cell surface expression of claudin 3 was determined by whole-cell indirect ELISA using chemiluminescence in Caco-2 cells exposed to sucralose or aspartame (0.1 mM) in the presence and absence of NAC. (**f**) ROS production was measured by fluorescence of DCFDA in Caco-2 cells transiently transfected with CLDN3-DYK or control vector, exposed to aspartame (0.1 mM). % permeability was calculated normalized to vehicle treatment. ROS production was calculated as % normalized to 0 µM. *n* = 5–6. Data are expressed as mean ± S.E.M. * *p* < 0.05 versus vehicle (0 µM); # *p* < 0.05 versus vehicle for NAC.

**Table 1 nutrients-12-01862-t001:** List of primers used for Caco-2 cell experiments. Taste receptor and claudin primers were purchased from QIAGEN, while the actin primers were purchased from Sigma-Aldrich.

Primer	Catalogue Number
T1R2	QT01026508
T2R3	QT00214270
Claudin 2	QT0089481
Claudin 3	QT00201376
Claudin 4	QT00241073
Claudin 7	QT00236061
Claudin 15	QT00202048
β-actin	Forward TCACCCTGAAGTACCCCATC Reverse TAGCACAGCCTGGATAGCAA

**Table 2 nutrients-12-01862-t002:** List of primers used for murine intestine claudin expression experiments. Primer sequences used were either published sequences or designed by Dr Rajagopal. The compiled list was kindly supply by Professor A Yu of the University of Kansas Medical Center.

Gene	Nucleotide Accession Number	Forward Primer	Reverse Primer	Reference
Claudin 1	NM_016674	CCTTCGGGAGCTCAGGTGCG	CCGCGTTGGCCATGGCTCTT	[31]
Claudin 2	NM_016675	TGGCGTCCAACTGGTGGGCT	ACCGCCGTCACAATGCTGGC	[31]
Claudin 3	NM_009902	GGGCAGTCTCTGTGCGAGCC	CGGACGTCTGTCGCCGGGAA	[31]
Claudin 4	BC132376	TGGGGACAGGCAAACCCGGA	CTTGCCGGCCGTAAGGAGCC	[31]
Claudin 5	NM_013805	GCTCAGTGCACCACCTGCGT	GAACCAGCAGAGCGGCACGA	[31]
Claudin 6	NM_018777	AGCACTCGCCCCCTCAACCTC	CATGGGCAGGGCACAGGACAC	[31]
Claudin 7	NM_001193619	CACGCAGAGCACCGGCATGA	AGGGCGAGCACCGAGTCGTA	Blast
Claudin 8	NM_018778	TCCCTGTCAGCTGGGTTGCCA	GCTCGCGCTTTAGGGCCACA	[31]
Claudin 9	NM_020293	TCCCAAGTGGCACCTCACGGT	CGCGTTCCTCTCTGCTGGCTG	[32]
Claudin 10a	NM_023878	GTGGCAGCAGGCAAGGCTGA	CACAGACGACGCTCGGGTGG	Blast
Claudin 10b	NM_001160099	CTCCATCTCGGGCTGGGTGC	CAACGCCAGCATGGAGGGGA	Blast
Claudin 11	NM_008770	TGGTTCCAGCTCGCCAACGC	TTACAGCACCTCGGCGGGCA	[32]
Claudin 12	NM_001193659	AGGTATTCCCGAGCGGAGCCA	CCCGGAGGCTTCAGGGAACCA	[32]
Claudin 13	NM_020504	TGACTCGTCCTGGTCCTGCCA	GGTCACCCTCCAAACGGGCA	[32]
Claudin 14	NM_001165926	GCAGCTGCGGCAAAGGAGTCT	ACGGCCGTCTAATGGGTCCCT	[32]
Claudin 15	NM_021719	TATGAACTGGGCCCCGCCCT	ATCCGAGGTGGCACGGGGTA	[32]
Claudin 16	NM_053241	CCACGAACCAGGATGTGCCCG	GCGAGGGTCGTGGAGGTCAC	[32]
Claudin 17	NM_181490	CTCCAGCGAGAGGGTCAAAG	AGCAGCAATATCCGCAGAGC	[32]
Claudin 18	NM_001194921	CCTGACACCAGATGACAGCA	GGCAACATTTTGGCCAGAGG	[32]
Claudin 19	NM_153105	CAGAGCCGGAGAGGGCGAACA	TCTGGGCAAGAGGGTTGCTGG	[32]
Claudin 20	NM_001101560	GCACTCTAAAATACTCCATTC	TGAAGCAGACTCCTCCAGC	Blast
Claudin 21		CTGGGACTATTGGGACTTCTG	AGGAGACTGGAAAGAGGGTAG	[33]
Claudin 22	NM_029383	TTCCGAACGGCAACGCAGGC	CCCATCCCAGCAGGGAGAGCA	Blast
Claudin 23	NM_027998	CGACGGACAGCATCGGCCTC	GGACTTGGGTGGCGGTCGTG	Blast
Claudin 24	NM_001111318	GAACGGCCATGCAATCAGTAGGGC	GACGCAGGATTTCCAGAGCCCC	Blast
Claudin 25	NM_171826	GAGAGGATGGGCGTATGCAG	ACTGCTCCAAGATGCTACGG	Blast
Claudin 26	NM_029070	GTGCGGGTGGGATCGCGTAA	CCCACGCTCCCCGTCTGTTC	Blast
Claudin 27	NM_001085535	TGGGTAGCCGGTGCCTCGAA	GCAGGCACCTAGCACAGGGG	[33]
β-actin	NM_007393	ATATCGCTGCGCTGGTCGTC	AGGATGGCGTGAGGGAGAGC	Blast

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
