# Peer review of "Artificial Sweeteners Disrupt Tight Junctions and Barrier Function in the Intestinal Epithelium through Activation of the Sweet Taste Receptor, T1R3"

_nutrients, 2020, doi:10.3390/nu12061862_

Round 1

Reviewer 1 Report

The manuscript presents very interesting study bringing us closer to final understanding of the issues related to intestinal permeability dysregulation. Overall, it is well written, introduction provides the good reason for conducting the study and leads smoothly towards the most complicated part of the text – the results section. Conclusions are fully supported by experimental data and the discussion is only very little speculative (lines 415-422).

The Introduction (line 33-34) unnecessarily limits the detrimental effect of increased intestinal permeability to “(…) entry of bacteria and bacterial proteins, such as lipopolysaccharide (…)” to sub-epithelial space and eventually to circulation. The less careful reader may overlook the fact that the experimental design of the study involves the LPS testing, and get the false impression that such “leak” in the intestines causes problems solely due to bacteria entering the tissues. Besides, this sentence should be rephrased because the LPS is not a protein, as the text suggests.

The lines 40-41 among the “(…) variety of treatments for impaired gut permeability (…)”, list antibiotics in the first place, which seems to be a little controversial. While some regulatory effect on the gut’s permeability exerted by SOME of the antibiotics may be suggested by the cited literature piece, there are many other reports pointing to antibiotics being responsible for dysbiosis and consequently the increased permeability.

The closing paragraph of the Introduction section (lines 79-90) instead of preparing the reader for the exciting discoveries awaiting for him in the Results section, seems to be rather a king of little summary of the findings. It should rather summarize the reasons for conducting your research.

Although the Results section is really exciting to read, it is supported by overwhelming amount of graphs, of which some might probably be combined together, others removed an substituted by brief description in the text. Reducing the number of graphs would highly improve the quality of this paper. Also, improving the quality of graphs (especially captions) would be of a great asset to this otherwise clear and scientifically important manuscript.

Author Response

Reviewer 1

The manuscript presents very interesting study bringing us closer to final understanding of the issues related to intestinal permeability dysregulation. Overall, it is well written, introduction provides the good reason for conducting the study and leads smoothly towards the most complicated part of the text – the results section. Conclusions are fully supported by experimental data and the discussion is only very little speculative (lines 415-422).

We thank the reviewer for these comments and have addressed their points as follows:

The Introduction (line 33-34) unnecessarily limits the detrimental effect of increased intestinal permeability to “(…) entry of bacteria and bacterial proteins, such as lipopolysaccharide (…)” to sub-epithelial space and eventually to circulation. The less careful reader may overlook the fact that the experimental design of the study involves the LPS testing, and get the false impression that such “leak” in the intestines causes problems solely due to bacteria entering the tissues. Besides, this sentence should be rephrased because the LPS is not a protein, as the text suggests.

We thank the reviewer for this comment and have now adjusted this statement to:

“Under normal conditions, the intestinal epithelial barrier maintains selective gut permeability to allow nutrient absorption but provide a robust barrier and prevent the entry of pathogens and pathogenic molecules into the circulation.”

The lines 40-41 among the “(…) variety of treatments for impaired gut permeability (…)”, list antibiotics in the first place, which seems to be a little controversial. While some regulatory effect on the gut’s permeability exerted by SOME of the antibiotics may be suggested by the cited literature piece, there are many other reports pointing to antibiotics being responsible for dysbiosis and consequently the increased permeability.

We agree that that the use of antibiotics is controversial and have now amended the sentence to remove antibiotics from the possible treatments for gut permeability.

The closing paragraph of the Introduction section (lines 79-90) instead of preparing the reader for the exciting discoveries awaiting for him in the Results section, seems to be rather a king of little summary of the findings. It should rather summarize the reasons for conducting your research.

We thank the reviewer for these comments and have amended the end of the Introduction to provide an overview of the aims of the project rather than a summary of findings:

“In the studies presented here, we use the well-established intestinal epithelial cell line, Caco-2, to study the effect of the commonly-consumed artificial sweeteners, aspartame, sucralose and saccharin, on intestinal epithelial cell viability, permeability and claudin expression, and oxidative stress response.  These studies utilised a variety of concentrations, which could be achieved in the diet, to address the impact of these sweeteners on the intestinal epithelium.  As artificial sweeteners stimulate the sweet taste receptors, T1R2 and T1R3, to elicit a taste response pathway [30], we also sought to demonstrate the role of these G-protein coupled receptors in regulating the effect of aspartame, sucralose and saccharin on the intestinal epithelium.”

Although the Results section is really exciting to read, it is supported by overwhelming amount of graphs, of which some might probably be combined together, others removed an substituted by brief description in the text. Reducing the number of graphs would highly improve the quality of this paper. Also, improving the quality of graphs (especially captions) would be of a great asset to this otherwise clear and scientifically important manuscript.

We appreciate these comments from the reviewer and would like to apologise for the issue with graphs in the manuscript. During uploading and generating the final merged version of the manuscript, there was a loss of clarity with the quality of graphs (including text visibility) which we believe has now been resolved.

We have further adjusted the number of graphs, to address these concerns, as follows:

  • From original manuscript, Figure 1 b (i) and (ii) – the mRNA and surface expression of T1R2 and T1R3 has now been removed and results have been included as text in the Results section (page 8). The Western blot data has been moved to Figure 1 b, thereby reducing the use of (i) and (ii) and simplifying Figure 1.
  • From original manuscript, Figure 2a, b and c (i) and (ii) – the permeability and viability data, following LPS treatment, is now included as text in the Results section (page 9) to simplify Figure 2.
  • From original manuscript, Figure 3a and b have been combined to form one graph which shows claudin expression of all segments from the small and large intestine all together. This has now simplified Figure 3.
  • From original manuscript, Figure 4a-e, the IgG control columns have now been removed from the bar graphs to simplify the figures.

Reviewer 2 Report

Overall the methods are very poorly described and more experiments need to be added to improve the impact of the paper.

MAJOR COMMENTS THAT NEED TO BE ADDRESSED:

-  The introduction

Introduction does not contain any information of T1R3 receptor.  The authors should add what is known of its role in physiology? What was the rationale in selecting T1R3 knockout as a model to study the effects of artificial sweeteners on gut integrity?

- Materials and methods:

It is unclear why the authors studied the expression of TJ proteins in mice. This not new information and it is useless if the authors do not study the effects of sweeteners on mice intestinal homeostasis.

The authors should add to the study at least a short term intervention with sweeteners in mice. Even stronger impact would be achieved by including a group of mice with T1R3 knockdown. Therefore, in its current form I consider this study as a nice preliminary study in cell cultures.

If the authors are not willing to include experiments in knockout animals, then the most important cell culture findings should be reproduced using another cell line, e.g. human colonic cell line NCM460.

Were the animals single-housed or co-housed? How many animals per cage? How was the circadian rhythm?

There is no information on how the cells were treated with artificial sweeteners. From where were the sweeteners purchased and how they were dissolved? What was used as a vehicle treatment?

Which % of T1R3 knockdown was achieved? The figure 1c shows only a very moderate reduction in T1R3. How many cells were plated for transfection? How much protein was loaded to SDS-Page?

Regarding ROS assay, what is DCFDA? How many cells were plated for the assay? What was used as a control treatment?

Why the cells were treated with LPS? In results it is mentioned as a positive control but the information should be given already in methods. And why the cells were not treated with a combination of artificial sweetener and LPS?

How was NAC dissolved and what was used as vehicle?

How many cells were plated for whole cell ELISA? How the results were normalized?

Which program was used for statistical analyses? Was the data normally distributed?

-  Results:

All figures, the concentrations used in the treatments should added to the graphs and not only in legends. The font size should be increased.

- Discussion:

The authors state in line 379 that “However, whether there is a direct effect of these sweet-taste molecules on intestinal permeability is not known”. But this is not true. Please correct and refer to e.g. a review published by Spencer et al. http://www.jnmjournal.org/journal/view.html?doi=10.5056/jnm15206

Similarly in line 382 the authors cannot state that they show the effects of artificial sweeteners for the first time on intestinal epithelial cells. The effects of saccharin on Caco-2 cells have already been studied: https://pubs.rsc.org/en/content/articlelanding/2018/FO/C8FO00883C#!divAbstract

MINOR SUGGESTED CORRECTIONS:

Line 50-53, the sentence should be divided in two sentences:

TJs are a family of around 50 proteins and, whilst the main TJ proteins are occludins, claudins and junctional-adhesion molecules (JAMs), knockout studies demonstrate a key role for claudins, rather than occludins and JAMs, in maintaining barrier integrity [10,11].

Line 77:

DO the authors really want to say gastric permeability? Should it be intestinal permeability?

Line 85:

permeability of what? Correct reactive oxygen species (ROS), because the abbreviation comes for the first time in the text. Then, in line 345 there is no need to write reactive oxygen species once it has been abbreviated

Author Response

Reviewer 2

Overall the methods are very poorly described and more experiments need to be added to improve the impact of the paper.

MAJOR COMMENTS THAT NEED TO BE ADDRESSED:

-  The introduction

Introduction does not contain any information of T1R3 receptor.  The authors should add what is known of its role in physiology? What was the rationale in selecting T1R3 knockout as a model to study the effects of artificial sweeteners on gut integrity?

 We thank the reviewer for these comments and have now included detail of T1R3 receptor in the Introduction (page 4) along with existing detail in the Results (page 8) and Discussion (page 13). The detail in the Introduction now describes that artificial sweeteners act by stimulating the G-protein-coupled receptor (T1R2/3), thus the importance of investigating the receptor in the study, for example:

“As artificial sweeteners stimulate the sweet taste receptors, T1R2 and T1R3, to elicit a taste response pathway [30], we also sought to demonstrate the role of these G-protein coupled receptors in regulating the effect of aspartame, sucralose and saccharin on the intestinal epithelium.” 

We further study T1R3 as the only sweet taste receptor which is identified in Caco-2 cells, by both our study (data in text, page 8) and O’Brien and Corpe (PloS One, 2016).

- Materials and methods:

It is unclear why the authors studied the expression of TJ proteins in mice. This not new information and it is useless if the authors do not study the effects of sweeteners on mice intestinal homeostasis.

We agree with the reviewer that this is a previously-studied area. We have therefore merged findings from the small and large intestine to provide one graph (Figure 3a) and altered the text in the Results section (page 10) to clarify that we are comparing how the profile of the most highly expressed claudins found in the murine gastrointestinal tract translates to the Caco-2 cell model:

“Although 26 claudins have been identified [37], in keeping with other studies [38,39] we demonstrate abundant expression of claudins 2, 3, 4, 7 and 15 using RT-PCR analysis of the small and large intestine of mice (Figure 3a).  We used this data to compare their expression levels in our Caco-2 cell model. ”

Our study demonstrates differences between claudin expression profile in the murine small and large intestine with that seen in Caco-2 cells. This finding allowed us to focus on claudins specifically relevant to the intestine and our study model.

The authors should add to the study at least a short term intervention with sweeteners in mice. Even stronger impact would be achieved by including a group of mice with T1R3 knockdown. Therefore, in its current form I consider this study as a nice preliminary study in cell cultures.

If the authors are not willing to include experiments in knockout animals, then the most important cell culture findings should be reproduced using another cell line, e.g. human colonic cell line NCM460.

We thank the reviewer for these comments and we agree that an in vivo study using T1R3 knockout mice would be a good follow on studyfrom the work presented here. We believe, however, that this would be a large-scale, lengthy and expensive study which is outside the scope of the current manuscript. While our findings are limited to one cell line, for mechanistic studies such as ours, there is significant evidence in the literature that Caco-2 cells are appropriate and relevant for studies on the intestine (e.g. Volpe et al, 2020, 15(5):539; Meunier V et al, 1995, 11(3-4):187; Bailey CA et al, 1996, 22(1-2):85).

Were the animals single-housed or co-housed? How many animals per cage? How was the circadian rhythm?

We thank the reviewer for this question and have now included this detail in the ‘Animals and Ethics’ section of the Methods (page 5) to state:

“Animals were housed in groups of 3-4 and were maintained on 12-hour light-dark cycling (7AM-7PM) at a temperature of 22-5 ºC”

There is no information on how the cells were treated with artificial sweeteners. From where were the sweeteners purchased and how they were dissolved? What was used as a vehicle treatment?

We thank the reviewer for these comments. The sweeteners were purchased from Sigma-Aldrich as per the ‘Cell lines and reagents’ section of the Materials and Methods section. The vehicle used for artificial sweeteners, (H2O), has now been defined and added to pages 6 and 7. The method of preparing the artificial sweeteners is now included on page 6:

Artificial sweeteners were dissolved in the vehicle control (H2O) and sterile-filtered to prepare a working stock solution. “

Which % of T1R3 knockdown was achieved? The figure 1c shows only a very moderate reduction in T1R3. How many cells were plated for transfection? How much protein was loaded to SDS-Page?

The values of T1R3 expression have been included in the text on page 9 (as below) to demonstrate the significant decrease in T1R3 expression in the transfected Caco-2 cells:

“We therefore next studied whether artificial sweeteners affected cell viability through T1R3, using siRNA knockdown of the sweet taste receptor (63.5 ± 2.7% decrease, p<0.05, n=6 (Figure 1b).”

Cell numbers plated were as per the manufacturer’s guidelines for the transfection reagents. This has now been included in the text of the Methods section (page 6):

“Cells were transfected at a seeding density of 0.5 x 104 cells per well of a 96 well plate, 2.5 x 104 cells per well of a Transwell insert, or 1.5 x 105 cells per well of a 6 well plate.”

Western blots were performed using 50 µg protein. This is now stated in the legend for Figure 1 and 5.

Regarding ROS assay, what is DCFDA? How many cells were plated for the assay? What was used as a control treatment?

We have now included detail on the DCFDA dye and explained, in full, the control treatment used (page 7). As below:

“Caco-2 cells were plated on black-walled 96-well plate for 24 h followed by exposure to the cell permeant, fluorogenic dye 2’,7’ –dichlorofluorescin diacetate (DCFDA) (10 µM), or DMSO control, for 30 min at 37oC in the dark. DCFDA was then removed and replaced with artificial sweeteners (range of concentrations), or vehicle control (H2O), or LPS (1 µg/ml) for 1.5 h. DCFDA fluorescence was measured at 488 nm using a fluorescent plate reader (Victor, Perkin Elmer) and measurements were compared to cells cultured in the absence of DCFDA.”

Why the cells were treated with LPS? In results it is mentioned as a positive control but the information should be given already in methods. And why the cells were not treated with a combination of artificial sweetener and LPS?

We thank the reviewer for these points and have now included that LPS was used as a positive control in the Methods section (pages 6 and 7), as well as the Results section.  As LPS was used as a model of permeability for the intestinal epithelial barrier studies, to compare against the permeability caused by artificial sweeteners, it was not thought necessary to perform any studies with combination of LPS and artificial sweetener. That is, the outcomes of the present study indicate damage to the epithelial monolayer similar to that seen in the positive control, LPS.

How was NAC dissolved and what was used as vehicle?

How many cells were plated for whole cell ELISA? How the results were normalized?

Which program was used for statistical analyses? Was the data normally distributed?

Thank you for these comments. The manuscript has been amended as follows to address these points:

  • vehicle for NAC (H2O) has now been included to two sections of the Method on page 6;
  • Both the whole cell ELISA and ROS assay methods have now been amended on page 7 to state -

“Caco-2 cells (1 x 104 cells per well) were plated on black-walled 96-well plate for 24 h…”;

  • Data for the whole cell ELISA are presented as mean ± SEM of relative fluorescence units data. Measurements from blank wells were subtracted to give the data presented. This has now been amended (page 7) to state -

“Fluorescent-conjugated secondary antibodies were measured at 1 sec exposure time using a florescent plate reader (Victor, Perkin Elmer) and measurements from blank wells (no primary antibody) were subtracted to provide the data presented.”

-  Results:

All figures, the concentrations used in the treatments should added to the graphs and not only in legends. The font size should be increased.

We appreciate these comments from the reviewer and would like to apologise for the issue with graphs in the manuscript. During uploading and generating the final version of the manuscript, there was a loss of clarity with the quality of graphs (including text visibility) which we believe has now been resolved.  The concentration for treatments is included in the legends for each figure however adding this detail to the graphs results in cluttered figures which lack clarity. Given the comments from reviewers 1 and 3, we are keen to keep the graphs as clear as possible.

- Discussion:

The authors state in line 379 that “However, whether there is a direct effect of these sweet-taste molecules on intestinal permeability is not known”. But this is not true. Please correct and refer to e.g. a review published by Spencer et al. http://www.jnmjournal.org/journal/view.html?doi=10.5056/jnm15206

Similarly in line 382 the authors cannot state that they show the effects of artificial sweeteners for the first time on intestinal epithelial cells. The effects of saccharin on Caco-2 cells have already been studied: https://pubs.rsc.org/en/content/articlelanding/2018/FO/C8FO00883C#!divAbstract

We thank the reviewer for these comments and agree that this systematic review (Spencer et al) and in vitro study (Santos et al) can be added to the manuscript.

The discussion now includes the following on page 14 to acknowledge previous studies –

“These studies are in contrast to previous findings where saccharin, but not aspartame or sucralose, was demonstrated to disrupt epithelial barrier integrity [59]. This difference in findings may be due to the short time point of 3.5 hour studied by Santos et al as oppose to the 24 hour time point assessed in the current studies.  Interestingly, in vivo studies demonstrate that a saccharin-enriched diet causes accumulation of water in the stool of rats which may be indicative of leak across the intestinal epithelium [60].” 

MINOR SUGGESTED CORRECTIONS:

Line 50-53, the sentence should be divided in two sentences:

TJs are a family of around 50 proteins and, whilst the main TJ proteins are occludins, claudins and junctional-adhesion molecules (JAMs), knockout studies demonstrate a key role for claudins, rather than occludins and JAMs, in maintaining barrier integrity [10,11].

We have now amended this sentence as follows on page 3:

“TJs are a family of around 50 proteins of which the main TJ proteins are occludins, claudins and junctional-adhesion molecules (JAMs) [9]. Despite this, knockout studies demonstrate a key role for claudins, rather than occludins and JAMs, in maintaining barrier integrity [10].”

Line 77:

Do the authors really want to say gastric permeability? Should it be intestinal permeability?

We thank the reviewer for this comment and agree that it should state ‘intestinal permeability’. This has now been adjusted.

Line 85:

permeability of what? Correct reactive oxygen species (ROS), because the abbreviation comes for the first time in the text. Then, in line 345 there is no need to write reactive oxygen species once it has been abbreviated

The final paragraph of the Introduction has now been amended as per Reviewer 1’s comments. We no longer describe reactive oxygen species here however we have adjusted line 345 to simply state ROS. We have also adjusted the terminology in the last paragraph of the introduction to specify “monolayer permeability” rather than just “permeability”.

Reviewer 3 Report

In this manuscript, Shil et al. studied the effects of artificial sweeteners on intestinal barrier functions in vitro. This paper could be important to public health since these compounds are widely incorporated into many food products and are consumed by the general public.

Some issues with the paper:

  • The figures are not clear, making it difficult to evaluate the data.
  • The concentration of the compounds for in vitro study is too high, and is not well justified.

In line 206-208, the authors stated “This range of concentrations (1000-10,000 μM) is achievable in the intestine, and thus physiologically-relevant, for members of the general population who regularly consume low-sugar foods, for example, one can of 208 soft drink contains up to 2 mM of artificial sweetener [24]”. However, the “2 mM” concentration is the concentration of artificial sweetener in beverage products, not the concentration in gut tissues.

Previous studies showed that after consumption, aspartame is rapidly, and completely metabolized into methanol, aspartic acid, and phenylalanine (Magnuson BA, Carakostas MC, Moore NH, Poulos SP, Renwick AG. Biological fate of low-calorie sweeteners. Nutr Rev 2016;74(11):670–89), and the parent compound, aspartame, does not reach the colon tissues. With this consideration, it is not feasible for the gut tissue to reach mM concentrations of these compounds. The authors need to better justify the doses used in the cell culture experiments.

  • In the Caco-2 permeability experiment, have the authors performed validation to make sure the cells have indeed formed a complete monolayer?

Author Response

Reviewer 3

In this manuscript, Shil et al. studied the effects of artificial sweeteners on intestinal barrier functions in vitro. This paper could be important to public health since these compounds are widely incorporated into many food products and are consumed by the general public.

Some issues with the paper:

  • The figures are not clear, making it difficult to evaluate the data.

We appreciate these comments from the reviewer and would like to apologise for the issue with graphs in the manuscript. During uploading and generating the final version of the manuscript, there was a loss of clarity with the quality of graphs (including text visibility) which we believe has now been resolved.

We have further adjusted the number of graphs, to address these concerns, as follows:

  • From original manuscript, Figure 1 b (i) and (ii) – the mRNA and surface expression of T1R2 and T1R3 has now been removed and results have been included as text form in the Results section (page 8). The Western blot data has been moved to Figure 1 b, thereby reducing the use of (i) and (ii) and simplifying Figure 1.
  • From original manuscript, Figure 2a, b and c (i) and (ii) – the permeability and viability data, following LPS treatment, is now included as text in the Results section (page 9) to simplify Figure 2.
  • From original manuscript, Figure 3a and b have been combined to form one graph which shows claudin expression of all segments from the small and large intestine all together. This has now simplified Figure 3.
  • From original manuscript, Figure 4a-e, the IgG control columns have now been removed from the bar graphs to simplify the figures.

  • The concentration of the compounds for in vitro study is too high, and is not well justified.

In line 206-208, the authors stated “This range of concentrations (1000-10,000 μM) is achievable in the intestine, and thus physiologically-relevant, for members of the general population who regularly consume low-sugar foods, for example, one can of 208 soft drink contains up to 2 mM of artificial sweetener [24]”. However, the “2 mM” concentration is the concentration of artificial sweetener in beverage products, not the concentration in gut tissues.

Previous studies showed that after consumption, aspartame is rapidly, and completely metabolized into methanol, aspartic acid, and phenylalanine (Magnuson BA, Carakostas MC, Moore NH, Poulos SP, Renwick AG. Biological fate of low-calorie sweeteners. Nutr Rev 2016;74(11):670–89), and the parent compound, aspartame, does not reach the colon tissues. With this consideration, it is not feasible for the gut tissue to reach mM concentrations of these compounds. The authors need to better justify the doses used in the cell culture experiments.

We thank the reviewer for these comments and have provided further detail and limitation in the justification of the concentrations used in the study on pages 8 and 9 –

“Whilst there are limited studies to indicate the concentration of sweeteners found in the intestine, following consumption of artificial sweeteners, this range of concentrations (1-10 mM) is potentially achievable in the intestine, and thus physiologically-relevant, for members of the general population who regularly consume significant amounts of artificially sweetened foods. For example, a single chewing gum contains 0.01 mM, one can of soft drink contains up to 2 mM of artificial sweetener and, more generally, the main additives in a range of products including diet drinks, sports drinks, snacks and confectionary are artificial sweeteners [23]. Given that the acceptable daily intake for these sweeteners is high (between 14-40 mg/kg body weight), it is likely that the public can consume high quantities of sweetener in the diet [23].”

  • In the Caco-2 permeability experiment, have the authors performed validation to make sure the cells have indeed formed a complete monolayer?

We thank the reviewer for this comment. Throughout our studies, we monitored the Caco-2 cell cultures closely, with brightfield microscopy, to ensure a complete intact and confluent monolayer. In addition, baseline levels of permeability, with both the TER measurement and FITC-dextran assay, demonstrate an intact epithelial monolayer. These values are in keeping with those published (e.g. Leonard M et al, 2000, 17(10):1181).

Round 2

Reviewer 2 Report

The authors have correctly answered to all my concerns and revised the manuscript. I do not have any further comments.

Author Response

We thank the Editor for these comments and have now expanded the discussion (page 14) to take these points into account.

The difference in exposure time between our study (24 hours) and the previous study by Santos et al (2 and 4 hours) is, we believe, linked to time-dependent changes in tight junction formation and thus leak across the epithelial barrier. We have included a justification of this in the discussion, citing supporting literature, to provide a discussion around why these differences in studies are observed.

We have also expanded on the point raised by the Editor regarding aspartame and physiological relevance of the study. This sweetener is hydrolysed in the latter regions of the small intestine. These segments of the small intestine would therefore be exposed to the hydrolysis products of aspartame (phenylalanine, aspartic acid and methanol). The earlier segments would, however, be more likely to be exposed to unmetabolised aspartame. Combined with the impact of T1R3 inhibition (siRNA) in blocking the effect of aspartame on Caco2 cells, we therefore believe that our findings have in vivo relevance.

We have also provided further justification of the concentrations used in the study. On page 9 we highlight that the concentrations used are below the ADI and on page 15 we relate the concentration used in our study to those in previous studies.

Reviewer 3 Report

The authors have adequately addressed all points that I requested. I do not have any further comments or suggestions.

Author Response

(The authors gave the same response as above.)
